# Non-Invasive Electric and Magnetic Brain Stimulation for the Treatment of Fibromyalgia

**DOI:** 10.3390/biomedicines11030954

**Published:** 2023-03-20

**Authors:** Andrés Molero-Chamizo, Michael A. Nitsche, Rafael Tomás Andújar Barroso, José R. Alameda Bailén, Jesús Carlos García Palomeque, Guadalupe Nathzidy Rivera-Urbina

**Affiliations:** 1Department of Clinical and Experimental Psychology, University of Huelva, Campus El Carmen, 21071 Huelva, Spain; 2Department of Psychology and Neurosciences, Leibniz Research Center for Working Environment and Human Factors, 44139 Dortmund, Germany; 3University Clinic of Psychiatry and Psychotherapy and University Clinic of Child and Adolescent Psychiatry and Psychotherapy, Protestant Hospital of Bethel Foundation, University Hospital OWL, Bielefeld University, 33615 Bielefeld, Germany; 4Histology Department, School of Medicine, Cadiz University, 11001 Cádiz, Spain; 5Cadiz Bahia Sur District, Andalusian Health Service, 11006 Cádiz, Spain; 6Psychology Center, Faculty of Administrative Sciences, Autonomous University of Baja California, Ensenada 22890, Mexico

**Keywords:** chronic pain, fibromyalgia, nervous system, neuromodulation, tDCS, TMS

## Abstract

Although fibromyalgia is defined by its core muscular nociceptive component, it also includes multiple dysfunctions that involve the musculoskeletal, gastrointestinal, immune, endocrine, as well as the central and peripheral nervous systems, amongst others. The pathogenic involvement of the nervous system and the numerous neurological and neuroinflammatory symptoms of this disease may benefit from neuromodulatory stimulation techniques that have been shown to be effective and safe in diverse nervous system pathologies. In this systematic review, we outline current evidence showing the potential of non-invasive brain stimulation techniques, such as therapeutic strategies in fibromyalgia. In addition, we evaluate the contribution of these tools to the exploration of the neurophysiological characteristics of fibromyalgia. Considering that the pathogenesis of this disease is unknown, these approaches do not aim to causally treat this syndrome, but to significantly reduce a range of key symptoms and thus improve the quality of life of the patients.

## 1. Introduction

The term fibromyalgia was introduced in 1976 [1] to replace the less precise terminology of fibromatosis that was previously used. The current criteria for the diagnosis of this syndrome are based on the definition of the American College of Rheumatology from 1990 (tender points or tenderness elicited by pressure of at least 11 of 18 specified body sites and widespread pain), which was updated by the American College of Rheumatology in 2010 and 2011 [2], and revised in 2016 [3] (generalized pain in at least four regions, symptoms present for at least 3 months, generalized pain index (GPI) ≥ 7, and severity of symptoms ≥ 5). Recent fibromyalgia prevalence studies estimate values between 0.2 and 6.6% in the general population, between 2.4 and 6.8 % in women, maximum values of 11.4% in urban areas, and 15% in special populations, such as car accident survivors [4].

Under this medical condition, a plethora of symptoms has been subsumed: chronic, widespread, and refractory pain, multiple tender points, continuous fatigue not associated with effort, allodynia, insomnia and other sleep disorders, depression, and anxiety as principal affective and emotional symptoms, cognitive impairment, paresthesia, allergic reactions and chemical and skin hypersensitivity, joint stiffness, tendinopathy, neuropathic and rheumatologic disturbances, irritable bowel syndrome, gastrointestinal disorders, headache, and several other poorly defined symptoms whose overlap has an unknown etiopathogenesis [5,6]. Because of this large number of overlapping symptoms, a multifactorial pathogenesis is discussed. Fibromyalgia thus seems to be an imprecise term, focused only on one of the multiple systems involved in the disease, and does not refer to its phenotype or its unknown etiopathogenesis [7]. This limited knowledge about the etiopathogenesis of the disease is one likely reason for the limited effectiveness of current symptomatic treatments, which have, at best, moderate therapeutic effects [6,8]. Until knowledge about the pathogenesis of this disease allows the development of a corrective causal treatment, the exploration of new effective and safe symptomatic therapeutic options is necessary to deal with the core symptoms of the disease.

In this systematic review, we focus on the involvement of the central nervous system in this syndrome and critically evaluate how non-invasive brain stimulation might be a useful method to explore the neurophysiological characteristics of fibromyalgia. Additionally, we highlight recent studies that point to the potential of non-invasive electric and magnetic brain stimulation [9,10] as a therapeutic approach to treat at least some of its neurological and neuroinflammatory symptoms [11], taking into account the apparent effectiveness of these methods to improve some physical indicators of health [12].

## 2. Methods

This review follows the Preferred Reporting Items for Systematic Reviews and Meta-Analyses (PRISMA) guidelines.

### 2.1. Search Strategy and Selection Criteria

Our search focused on research studies about neurophysiological alterations in fibromyalgia as evaluated by transcranial magnetic stimulation, as well as studies investigating the effects of non-invasive magnetic or electric brain stimulation methods on fibromyalgia symptoms. We conducted a systematic search in PubMed and Web of Science databases using the following keyword combinations: “transcranial magnetic stimulation” OR “TMS” OR “rTMS” AND “fibromyalgia” OR “transcranial electric stimulation” OR “tES” AND “fibromyalgia” OR “transcranial direct current stimulation” OR “tDCS” AND “fibromyalgia” OR “transcranial alternating current stimulation” OR “tACS” AND “fibromyalgia” OR “transcranial random noise stimulation” OR “tRNS” AND “fibromyalgia” OR “transcutaneous vagus nerve stimulation” OR “tVNS” AND “fibromyalgia”. The database search included publications from 2000 until January 2023. This search yielded a total of 572 results, as shown in the PRISMA flow diagram of Figure 1.

We selected research articles that met the following criteria: adult participants with fibromyalgia, studies with non-invasive electric and/or magnetic brain stimulation (open-label, single or double-blind, with or without sham condition, and healthy participants, randomized or not randomized studies). The search was restricted to full-text original articles published in English. Fifty-four studies met these criteria and were included in this systematic review (Figure 1). Review and study protocol articles were only reported in the Discussion Section or to support general concepts, but they were not considered as research results.

### 2.2. Data Extraction and Analysis

We analyzed the following information from the full-text articles: study population (fibromyalgia patients, patients with other clinical conditions, healthy controls), sample size (per group and total), study design (between-subjects with parallel groups, within subject-crossover, single case study, single- or double-blind, open-label study, randomized study, sham-controlled, longitudinal study), TMS-dependent neurophysiological measures of cortical excitability, TMS and tDCS protocols and stimulation parameters, cortical stimulation target, tDCS electrode position, the total number of stimulation sessions and duration, sessions per week, pain measures, questionnaires and other outcome measures, tasks, physiological measures, results reported, and adverse effects.

## 3. Results

The results of the selected studies were organized according to the stimulation method (TMS, rTMS, and tDCS). These data are shown in Table 1, Table 2 and Table 3.

### 3.1. Nervous System Involvement in Fibromyalgia

Several of the multiple symptoms of fibromyalgia have an evident neurological contribution, such as headache, irritable bowel syndrome, allodynia, sleep disorders, fatigue, depression, anxiety, post-traumatic stress disorder, cognitive impairment, paresthesia, and neuropathic pain [62]. The last probably includes chronic central sensitization, a maladaptive plasticity mechanism of the brain in response to chronic pain [63]. This sensitization results in increased responsiveness to all kinds of stimuli with pain perception. Probable changes in gray matter volumes of the pain brain matrix have been shown in fibromyalgia patients by magnetic resonance imaging (MRI) [64], but evidence for systematic structural and functional alterations in the pain processing network is still limited. Likewise, the functional integrity of the descending pain-modulating system seems to be compromised in these patients [65], which could be a target for therapeutic interventions. In recent years, chronic pain in fibromyalgia has also been related to abnormal functioning of the C-fibers of spinal neurons of the dorsal horn [66,67], which may contribute to central sensitization as well [63]. Via transcranial magnetic stimulation (TMS), a distinctive dysfunction of intracortical excitation and inhibition processes has been identified in basic and clinical fibromyalgia studies (see below in the TMS section). This altered cortical activity pattern includes decreased intracortical inhibition and facilitation in the primary motor cortex (M1), and this dysfunctional excitability pattern suggests an altered basal neuronal plasticity state in the cerebral motor network that might be associated with some clinical symptoms modulated by the sensorimotor system, such as chronic and widespread pain, probably strengthened by maladaptive LTP-like plasticity of the pain matrix [13].

A possible direct or indirect involvement of the nervous system in the pathogenesis of fibromyalgia symptoms is also suggested by the fact that widespread pain in these patients may be relieved via psychoactive drugs with different pharmacodynamic properties, such as antidepressants (duloxetine, milnacipran, tricyclics), anxiolytics (benzodiazepines), muscle relaxants (cyclobenzaprine), and anticonvulsants (pregabalin) [68,69,70,71], which are however not analgesics in the strictest sense. The combination of neurological, psychological, and neuropsychiatric symptoms, as well as the moderate effects of available psychopharmacological therapies in fibromyalgia, encourage the further development of central nervous system interventions as adjuvant therapies in the treatment of this complex disease. Non-invasive brain stimulation could be one such intervention, considering its advantageous safety profile and effectiveness in the treatment of several neurological, neuropsychological, and neuropsychiatric disorders [72,73], including pain relief [74]. Moreover, brain stimulation with non-invasive methods is a useful tool for studying the cortical activity of fibromyalgia patients in basic studies [67], which could provide potential neurophysiological targets for neuromodulation therapies. In summary, evidence indicates that some fibromyalgia symptoms involve central nervous system mechanisms, mainly cortical plasticity alterations and central sensitization, and these perturbances of the nervous system may be tackled by non-invasive brain stimulation methods.

According to this background, in the following sections we show how non-invasive brain stimulation, particularly specific TMS protocols, may help to understand the pattern of cortical excitation and inhibition in fibromyalgia patients. We also analyze the level of evidence for the potential effectiveness of commonly used non-invasive brain stimulation methods to treat some of the most relevant nervous system-dependent fibromyalgia symptoms.

### 3.2. Contributions of Non-Invasive Brain Stimulation

The two most frequently used non-invasive brain stimulation tools in fibromyalgia research (for both clinical and basic studies) are transcranial magnetic and electrical stimulation. Both approaches include different variants (i.e., with single, double, or repetitive magnetic pulses or with direct, alternating, random oscillatory, or pulsed current, to mention the most relevant magnetic and electric stimulation procedures), which are applied with different stimulation parameters (intensity, frequency, duration, etc.) and protocols (number of sessions, target area, in combination with other interventions, etc.).

Single and double-pulse TMS protocols are applied not for therapeutic purposes but for monitoring disease-related cortical excitability alterations and, thus, exploring pathophysiological features of fibromyalgia [14]. These tools may therefore facilitate the design of neuromodulation strategies aimed at minimizing or antagonizing possible dysfunctions of cortical excitability. For therapeutic purposes in the context of fibromyalgia, most clinical evidence is available for the application of repetitive transcranial magnetic stimulation (rTMS) and transcranial direct current stimulation (tDCS), both showing therapeutic effects with a moderate level of evidence [9,75]. rTMS consists of the application of repetitive magnetic pulses over the scalp with different frequencies. This TMS variant modulates the cortical activity and excitability beyond the end of the stimulation period, and specific stimulation frequencies induce excitability enhancement or reduction. Underlying mechanisms involve long-term potentiation (LTP) and long-term depression (LTD)-like plasticity processes. tDCS, on the other hand, consists of the application of weak direct currents (in the mA range) for a few minutes via electrodes (anode and cathode) placed on the scalp, which promotes polarity-dependent cortical excitability changes via the induction of subthreshold alterations of the resting membrane potential, and induction of LTP- and LTD-like plasticity. Under conventional protocols, anodal tDCS increases cortical excitability and cathodal tDCS reduces it. Some stimulation parameters, such as intensity and duration, may influence these effects in a non-linear way. Moreover, specific tDCS protocols, particularly those involving multiple sessions, have also shown the potential to induce prolonged neuroplastic and behavioral effects [73].

Transcranial alternating current stimulation (tACS) applies oscillatory sinusoidal, not tonic currents, with similar current strengths as tDCS, that entrain brain oscillations. It has been relatively rarely explored as a therapeutic tool to treat central nervous system activity-dependent fibromyalgia symptoms. Only a recent study has reported effects so far [76], thus, evidence for its efficacy is limited at present [76,77]. Other non-invasive electrical brain stimulation techniques, such as transcranial random noise stimulation (tRNS), have also been explored in the context of fibromyalgia, but their effectiveness is unclear.

In the following sections, we provide details about the specific stimulation parameters and protocols applied via magnetic and electric brain stimulation methods, both in clinical and basic fibromyalgia studies, to present currently available knowledge about the usefulness of these brain stimulation techniques in the study of the neurobiology of fibromyalgia, and in the clinical context of this disease.

### 3.3. Transcranial Magnetic Stimulation

We identified fifteen studies showing either the clinical effects of rTMS on different fibromyalgia symptoms or a distinctive neurophysiological pattern of these patients as revealed by single and double-pulse TMS. Stimulation parameters, specific protocols, outcome measures, sample sizes, and evaluated symptoms are somehow heterogeneous between studies, which limits their comparability. Table 1 (TMS) and Table 2 (rTMS) summarize the principal protocols and results of these studies [13,14,15,16,17,18,19,20,21,22,23,24,25,26]. Overall findings for TMS and rTMS are described in the following subsections.

#### 3.3.1. Single and Double Pulse TMS

Single and double-pulse TMS protocols are useful for exploring differences in cortical excitability between fibromyalgia patients, other pathologies, and healthy humans (Table 1). Particularly, TMS protocols tackling M1 as a model system for human cortical excitability indicate an alteration of intracortical inhibition [13], involving mainly reduced short-latency intracortical inhibition (SICI) in fibromyalgia patients, which suggests central disinhibition mechanisms in this syndrome [14,18,19]. However, reduced SICI has not been demonstrated in all studies [17]. A lower inhibitory control of the descending pain modulatory system (DPMS) was also revealed in fibromyalgia patients, in accordance with the observed reduced intracortical inhibition [17,20]. These disinhibition processes are related to higher levels of plasma BDNF [17,18,20]. On the other hand, reduced intracortical facilitation (ICF) has also been described in fibromyalgia [14], but this finding has not been as consistent as that observed with inhibitory processes. Although the specific pathophysiology of fibromyalgia that affects cortical excitability is not well understood [16], empirical results show that M1 stimulation via rTMS reduces pain intensity and modulates intracortical inhibition processes [25]. Further studies are needed to elucidate whether these cortical excitability alterations are a primary central mechanism of fibromyalgia and its relationship to chronic pain and other symptoms.

#### 3.3.2. rTMS

The second general finding of TMS studies is related to the potential therapeutic effects of rTMS in fibromyalgia. Different rTMS protocols have been explored in clinical trials, and the results reveal moderate positive effects on pain, quality of life, anxiety, and depression [21,23,24,25,26], as well as improvements of several other symptoms, such as catastrophizing [22], general activity and sleep disturbances [25], and physical and general fatigue [23]. All these were randomized, double-blind, and sham-controlled studies, except one, in which randomization was not reported [26], and targeted the left M1 or left DLPFC, with the exception of a single study targeting the dorsal anterior cingulate cortex (dACC) [26]. The number of sessions ranged between 10 and 16 when targeting M1 and between 10 and 20 when stimulating the DLPFC. The total sample size was between 30 and 49 in the M1 studies and between 20 and 26 in the DLPFC studies. The dACC study included a sample of 32 participants (Table 2).

Mechanistically, the long-lasting analgesic effect of rTMS in fibromyalgia has been linked to the stabilization of cortical inhibition and facilitation. Active, but not sham, facilitatory rTMS with 10 Hz over M1 has been shown to increase SICI and ICF, and reduce pain intensity in fibromyalgia patients [25]. This relation between improved intracortical excitability measures and reduced pain might be mediated by a re-establishment of the dysfunctional descending pain control system [17,20]. Alternatively, the analgesic effects of rTMS might depend on the inhibition of thalamic sensory nuclei induced by motor nuclei activation. The rTMS protocol with the best evidence with respect to its analgesic potential is high-frequency rTMS (with a frequency of 10 Hz) applied over the left M1 (80% intensity of the resting motor threshold, RMT) [21,25], which is also associated with significant improvements in the quality of life measures [22]. The analgesic effects of this protocol are moderate to high and are not limited to the intervention period, although the heterogeneity of the measured follow-up periods of the different studies makes definitive conclusions about the exact stability of effects difficult. rTMS (also with 10 Hz trains) over the left dorsolateral prefrontal cortex (DLPFC) at 120% RMT has shown moderate effects on pain intensity, fatigue, and depression, that continue for several weeks during the follow-up period [23,24], and 10 Hz rTMS, but not 1 Hz rTMS, at 110% of RMT over the dorsal anterior cingulate cortex (dACC) also induced strong pain intensity reductions lasting for 4 weeks after the last session [26]. In all of these studies, reported adverse effects were minor and transient (headache, neck pain, flashes, dizziness, nausea) (Table 2).

#### 3.3.3. Transcranial Direct Current Stimulation

We identified thirty-five studies reporting the use of tDCS to improve fibromyalgia symptoms (mainly pain intensity), and some of these studies correlated symptomatic improvements with changes in neurobiological measures. The specific protocols followed in each study and the respective results are shown in Table 3 [27,28,29,30,31,32,33,34,35,36,37,38,39,40,41,42,43,44,45,46,47,48,49,50,51,52,53,54,55,56,57,58,59,60,61].

There is no consensus about the optimal tDCS protocol to treat pain and other fibromyalgia symptoms. However, all studies reported significant effects of particular tDCS configurations on different fibromyalgia symptoms (from moderate to high), thus highlighting the potential of this non-invasive brain stimulation method to improve a variety of symptoms. Overall, these studies reveal that anodal tDCS applied over the left M1 (1/2 mA, 20 min, 1/5/10/15 sessions) [42,44,47,48,51,52], and to a lesser extent, over the DLPFC (1.5/2 mA, 20/30 min, 3/8/10/20/60 sessions) [27,31,34,36,39,41], has therapeutic effects on pain, with moderate to high levels of effectivity. With two exceptions [47,51] in the case of M1, and four in the case of the DLPFC stimulation [27,31,36,39], these studies were randomized, double-blind, and sham-controlled. The number of sessions of M1 tDCS ranged between 1 and 15, with 5 sessions being the most frequently used protocol. When targeting the DLPFC, the number of tDCS sessions ranged between 3 and 60, with greater variability in the number of sessions in this case, and without a common pattern of repetitions in these studies. The total sample size in the M1 studies was between 12 and 36, and between 20 and 58 in the DLPFC studies (Table 3).

In addition to measures of chronic pain, these studies assessed other characteristic symptoms of the disease, such as fatigue, quality-of-life related to pain, sleep disturbances, and cognitive and emotional functions. tDCS over M1 induced significant effects on these symptoms as well, although with a lower degree of effectiveness and evidence in relation to pain improvements [36]. Moreover, when symptomatic changes induced by anodal M1/DLPFC tDCS were contrasted with pre- and post-intervention physiological measures, some associations were found. Increased intracortical excitability as indexed by TMS, increased cortical activity as monitored by functional MRI (fMRI) in the rostral anterior cingulate cortex/ventromedial prefrontal cortex (rACC/vmPFC), reduced activity in the posterior insula, reduced serum BDNF and increased beta-endorphin levels, among other biological measures, were positively correlated with pain improvements [30,38,39,41,42]. No serious adverse effects were reported. Frequently reported side effects were the typical transient effects associated with tDCS, i.e., tingling, slight burning, and skin redness beneath the stimulation electrode.

In contrast to these largely consistent results, one study reported no significant effects of tDCS [49], as compared with the sham condition, on specific fibromyalgia symptoms, such as pain, fatigue, cognitive and sleep disturbances, and hyperalgesia. This double-blind study with four treatment groups (20 min of anodal tDCS, 2 mA stimulation intensity) conducted the intervention over the left M1, DLPFC, or the operculo-insular cortex, vs. sham stimulation, over 15 sessions, for 5 consecutive days per week. The return electrode was positioned over a typical contralateral region, the supraorbital area (Fp2 EEG electrode position), which is the most frequently used electrode placement for the cathodal return electrode in fibromyalgia studies. During the trial, the patients kept the usual medication of the two previous months. The primary outcome was pain intensity, and secondary outcomes were fatigue, mood, cognitive and sleep disturbances, and hyperalgesia measured by pressure pain threshold. Measurements were taken at 3-time points (before, immediately after treatment, and at 6 months follow-up). Thus, the tDCS parameters (intensity and density of current, session duration, number of sessions, etc.) and protocol in this study were comparable with those implemented in other studies, which reported significant effects of the intervention on these and other symptoms. Tentative explanations for this single negative effect may include the well-known inter-individual variability of tDCS effects (dependent on anatomical, physiological, behavioral, psychological, and other characteristics). Interestingly, the three active tDCS groups (anodal stimulation over the left M1, left DLPFC, and left operculo-insular cortex) showed a significantly larger improvement in the secondary outcome measures of anxiety and depression, compared with the sham condition, which reveals the effectiveness of tDCS at least for the treatment of affective symptoms in fibromyalgia also in this study. Taken together, evidence indicates that conventional protocols of anodal tDCS over the left M1 and DLPFC are an effective and safe neuromodulation approach with moderate effects on the core symptoms (pain, fatigue, poor quality of life, cognitive and mood alterations) of fibromyalgia.

#### 3.3.4. Other Non-Invasive Neuromodulation Methods

The effects of other non-invasive neuromodulation techniques on fibromyalgia symptoms have been explored to a lesser extent. Despite limited evidence for the effectiveness of transcutaneous electrical nerve stimulation (TENS) [78] and transcutaneous spinal cord stimulation (tSCS) for fibromyalgia [79] or peripheral neuropathic pain symptoms [80], other approaches seem to provide more promising results.

Since the parasympathetic vagus nerve is involved in neurovegetative and inflammatory processes, this cranial nerve is considered a target for neuromodulation in fibromyalgia patients. Invasive [81] and non-invasive [82,83,84] methods of vagal nerve stimulation have been shown to be effective in different neurological disorders. In this vein, non-invasive transcutaneous vagus nerve stimulation (tVNS) is a promising therapeutic approach because of its safety and efficacy profile [85]. Particularly, transcutaneous stimulation of the auricular and cervical branches of the vagal nerve induces modulatory effects on the activity of this nerve [86,87], with therapeutic impact in several pathologies [86,88]. Thus, the sympathetic and inflammatory dysregulation that appears to underlie fibromyalgia might be tackled non-invasively by tVNS [85], and some studies indeed suggest therapeutic effects of tVNS on pain and quality of life in patients with fibromyalgia [84,89].

Other attempts to treat fibromyalgia symptoms by neuromodulation were motivated by the observation of abnormal slow cortical alpha oscillations in these patients. For modulating such altered oscillations, the effects of tACS (35 cm^2^ electrode size, intensity ranging from 1 to 2 mA, 30 min per session, 10 sessions, 5 times a week), followed by 60 min of physical rehabilitative exercise, was evaluated in a randomized, crossover (with a washout period of 4 weeks) study [76]. Patients showing a higher spectral power of low frequencies at baseline (theta, delta, alpha1) were stimulated with beta-tACS at 30 Hz, and patients with a higher spectral power of fast frequencies (beta, alpha2) were stimulated with theta-tACS at 4 Hz. In each patient, one electrode was positioned over the scalp area showing the highest spectral power difference with respect to healthy controls, and the other electrode was positioned over the ipsilateral mastoid. The effects on pain and cognitive symptoms were compared in all patients with those induced by tRNS, a non-invasive electric brain stimulation technique in which random oscillations are applied in a frequency band up to 600 Hz. The tRNS parameters set in this study were: alternating current with random amplitude and frequency, within the range of 1–2 mA and 0–100 Hz, respectively. Despite this a priori likely ineffective tRNS protocol, both methods improved symptoms after interventions, but, since this study did not include a conventional sham control, these results are of limited value [77]. However, preliminary data suggest that a stimulation protocol with 10 sessions (5 consecutive days per week for 2 weeks) of tRNS (electrodes positioned over the M1 and the right supra-orbital regions, 1.5 mA current intensity randomly oscillating in the frequency range of 101–640 Hz, for 10 min per session) is an effective approach to improve pain and Fibromyalgia Impact Questionnaire (FIQ) scores, as well as depression and anxiety symptoms in fibromyalgia patients [90]. This protocol also resulted in improved cognitive symptoms, which suggests that the application of randomly changing alternating currents might be an effective method to modulate cortical excitability via the induction of electric fields less dependent on neural orientation, as compared to the tonic fields induced by tDCS [90].

## 4. Discussion

In this systematic review, we highlight the usefulness of single and double-pulse TMS to explore the neurophysiological dynamics of fibromyalgia, particularly with respect to cortical excitatory and inhibitory activity, and we critically review the therapeutic application of two main interventions in this field, rTMS and tDCS. We also present the first results of emerging tES protocols, such as tACS and tRNS.

### 4.1. TMS and Cortical Neurophysiology in Fibromyalgia

Regarding exploration of the neurophysiological pattern of fibromyalgia, single and double pulse TMS has shown to be a useful tool to explore intracortical and cortico-spinal excitability characteristics of patients, which may help to define a profile of central nervous system biomarkers of this disease [14,91]. One such biomarker identified in fibromyalgia is reduced intracortical inhibition, particularly SICI, as indexed by TMS measures. This cortical disinhibition is also related to dysfunctions in the descending pain-modulating system [75]. Cortical excitability alterations in fibromyalgia are, however, not limited to reduced SICI, since reduced ICF has also been reported [13,14]. Different non-invasive brain stimulation methods with the potential to modulate cortical excitability have been explored in clinical trials with fibromyalgia patients. The results of these studies reveal that rTMS and tDCS indeed induce neuromodulatory effects and, through neural mechanisms not yet fully understood, reduce pain intensity in these patients [25,91,92].

### 4.2. Therapeutic Effects of Non-Invasive Brain Stimulation on Fibromyalgia Symptoms

The exploration of new approaches for the treatment of fibromyalgia is important, due to the limited effectiveness of current pharmacological and physical therapies to face the symptoms of this syndrome. Although the etiopathogenesis of fibromyalgia is unknown, some critical symptoms are apparently related to dysfunctions of the nervous system. These alterations could potentially be tackled by different techniques of non-invasive neuromodulation. The results of the reviewed clinical studies reveal that electric and magnetic brain stimulation are safe neuromodulation methods (with slight and transient side effects) for the treatment of fibromyalgia, and they suggest that their overall effectiveness in improving pain intensity, fatigue, and quality of life is moderate (Table 2 and Table 3). As brain stimulation targets, M1 and DLPFC (mainly over the left hemisphere) had similar pain-reducing effects in separate studies, although the available evidence for therapeutic potential is somewhat stronger in the case of M1 stimulation.

#### 4.2.1. tDCS

Repetitive sessions of anodal tDCS applied over the left M1 have shown to be effective in significantly reducing pain intensity when compared to sham [29,30,33,38,39,40,42,47,52,53,54]. Meta-analyses of respective clinical studies came to similar results on pain and also described beneficial effects on affective and cognitive symptoms and quality of life [92,93,94]. The partially heterogeneous effects shown in these studies [11] may be due to the heterogeneity of experimental designs (number of tDCS sessions, crossover studies, parallel groups, follow-up periods, and others), protocols (current intensity, additional interventions, and others), and outcome measures (VAS, HPT, VNS, LF-MPQ, SF-MPQ, NRS, PCP-S, PPT, WPI, and others) (Table 3). The main evidence for the effectiveness of tDCS in fibromyalgia is based on protocols that include multiple sessions (five as the most frequently applied number of sessions) of anodal tDCS applied over the left M1, under conventional current intensities and stimulation duration (1/2 mA, 20 min) [42,47]. Most of these studies, but not all, included randomization, double-blinding and a sham control. Similar protocols targeting the DLPFC also show significant therapeutic effects on pain and emotions, although the effect size here is medium overall [44]. All analyzed studies showed a good safety profile of tDCS in fibromyalgia patients, regardless of the number of sessions, current intensity, and cortical target (Table 3). Other non-invasive electric stimulation tools, such as tVNS, tACS, and tRNS, are currently being explored as potential therapeutic approaches in fibromyalgia. Evidence for a relevant treatment effect of these protocols is very limited at present [95], although preliminary results are similarly promising.

#### 4.2.2. rTMS

As with anodal tDCS, excitatory stimulation with rTMS at 10 Hz over the left M1 significantly reduces pain intensity [25], improves cognitive and emotional functions [21], and impacts on some physiological measures [22]. Although the exact mechanism is unknown, the analgesic effects of sensory-motor non-invasive electrical and magnetic brain stimulation might be related to neuromodulatory actions on pain processing [75]. According to the proposed mechanisms of non-invasive brain stimulation in chronic pain control [96,97], the analgesic effects of these methods on fibromyalgia patients may well depend on the modulation of thalamic-cortical connectivity and resulting pain perception reduction. Complementarily, a somatosensory cortex inhibition triggered by M1 stimulation might also contribute to pain reduction in these patients. Consistent with these hypothesized mechanisms, recent meta-analyses support the effectiveness of high-frequency rTMS over the left M1 to reduce fibromyalgia-related pain after the intervention, with short- and long-term therapeutic effects [9,98]. Similar rTMS protocols over the DLPFC are also effective and safe for treating pain in these patients [34]. The effects on pain reported after DLPFC stimulation might depend on its potential to regulate the emotional evaluation of pain [95]. The design of the clinical rTMS studies targeting M1 or DLPFC followed quality standards, including randomization, double-blinding, and sham-control, and included more than ten stimulation sessions and medium sample size in general. Fatigue and emotional symptoms have also been addressed in a few of these studies [23], but evidence to determine which rTMS protocols might be most effective and which cortical areas should be targeted for the treatment of these symptoms is missing. Moreover, similar to tDCS, rTMS-related beneficial effects on chronic pain not caused by fibromyalgia have also been described, and this intervention has been shown to be safe for pain treatment in general [99]. This TMS method is usually associated with minimal side effects, mainly headache and stimulation discomforts, all of them transient.

### 4.3. Future Directions and Limitations

With regard to future directions of research, the available data are promising regarding the possibility of implementing non-invasive brain stimulation protocols in the treatment of, at least, pain symptoms in patients with fibromyalgia. However, it is still necessary to develop standardized stimulation protocols that induce longer-lasting effects and address various symptoms, in this case perhaps, through specific procedures based on the cortical target with the greatest clinical evidence (M1 and DLPDFC), the optimal stimulation method (tDCS or rTMS) and protocol, and personalization of treatment. Potential interactions with other therapies, including pharmacological treatment, are uncertain, and due to the impact these may have on the effects of brain stimulation, these interactions should be systematically analyzed in future research. On the other hand, non-invasive brain stimulation approaches in the clinical context require considerable expertise and assistance in the hospital setting, which makes access to these therapies difficult at present for certain treatments. In this connection, tailored stimulation through home-based tDCS might facilitate the treatment of symptoms related to the central nervous system in this disease [41], although this potential advantage also needs to be systematically explored.

## 5. Conclusions

Non-invasive neuromodulation procedures, particularly rTMS and tDCS, have developed as effective and safe alternatives, or adjuvant treatments, for patients with fibromyalgia. However, so far, proof of concept studies have been primarily conducted, and protocols have to be systematically explored in future studies to improve the efficacy and feasibility of these methods for the clinical therapy of fibromyalgia. Firstly, stimulation protocols should be standardized regarding optimal dosage, number of sessions, and brain target. Secondly, interindividual variability in response to brain stimulation might require personalized procedures, which also should be tailored to specific symptoms, considering the wide range of dysfunctions in fibromyalgia. This personalization of treatment may help to optimize respective interventions and thus promote the maintenance of therapeutic effects over time. Thirdly, interactions with other therapies and their possible adjunctive effects are unknown, and this is one of the critical factors to be investigated due to the complexity of this disease. Fibromyalgia patients are usually under polypharmacotherapy [100], and this makes it difficult to analyze the drug interactions in each case with the effects of different techniques and stimulation protocols.

Although non-invasive brain stimulation appears to be effective for the treatment of fibromyalgia, rTMS and tDCS effects might not be specific to the symptoms of this pathology. Thus, rTMS protocols described in clinical fibromyalgia studies have shown similar effectiveness for relieving rheumatic [101,102] and post-stroke [103] pain. Likewise, the therapeutic effects of electric brain stimulation also do not seem to be restricted to fibromyalgia, as pain improvements after tDCS interventions have been described in patients with different pain syndromes (trigeminal neuralgia, poststroke pain syndrome, back pain, and fibromyalgia) [104]. This suggests that the clinical improvement induced by non-invasive neuromodulation is at present mainly due to a primary effect on central pain processing, which is also affected in other chronic pain pathologies.

## Figures and Tables

**Figure 1 biomedicines-11-00954-f001:**
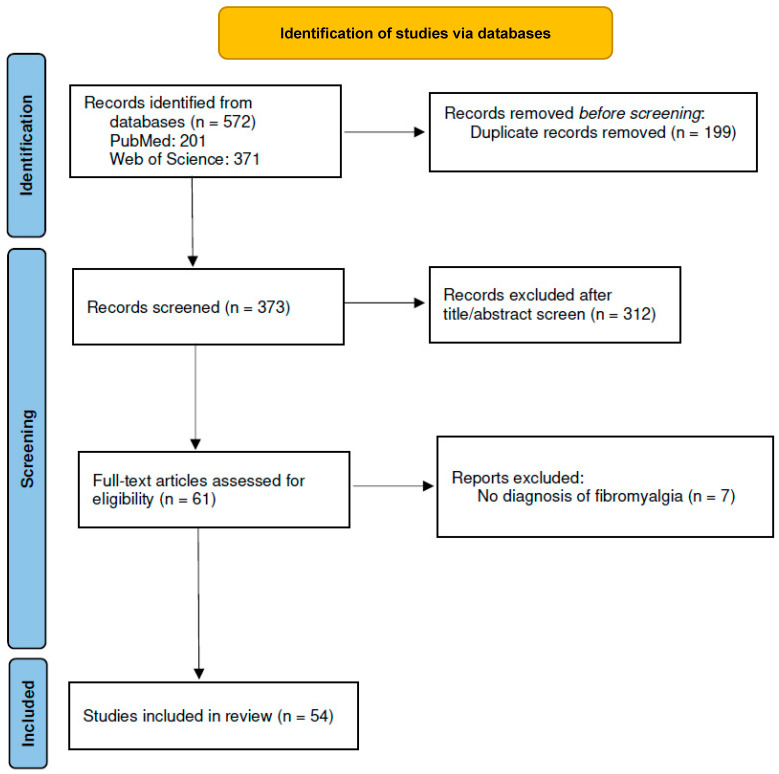
Preferred Reporting Items for Systematic Reviews and Meta-Analyses (PRISMA) flow diagram of included studies (408422).

**Table 1 biomedicines-11-00954-t001:** Single- and double-pulse TMS protocols and results of relevant studies conducted to explore the neurobiological features of fibromyalgia patients.

Reference	Sample Size	TMS Measures	Target Area	Pain Measures	Other Measures	TMS Results	Other Results	Adverse Events
[13] Salerno et al., 2000	13 FM W, 13 HC W, 5 RA W	spTMS: CSP, MEPs ppTMS: SICI, ICF	M1	None	None	FM: ICF↓, SICI↓, CSP↓	None	NR
[14] Mhalla et al., 2010	46 FM W, 21 non-FM W	spTMS: RMT, MEPs ppTMS: SICI, ICF	M1	VAS	CPM, EEG, fatigue, anxiety, depression, CAT	FM: ICF↓, SICI↓	r- between ICF/SICI and fatigue, depression, and CAT scores	NR
[15] Uygur-Kucukseymen et al., 2020	36 FM (23 W)	spTMS: MEPs ppTMS: SICI, ICF	M1	VAS	CPM, EEG	r+ between SICI and theta ERD; r- between ICF and delta ERD	r- between VAS scores and α/ß EEG power; r- between VAS and θ/δ power. r+ between ERD and δ power	NR
[16] Tiwari et al., 2021	34 FM (W), 30 PFC (W)	spTMS: RMT, MEPs	M1	NPRS	MMSI, Stroop color-word, ESS, PSQI	FM vs. PFC: No TMS differences	FM vs. PFC: MMSI↓, Stroop↓, RT↑, PSQI↑	Occasional mild headache (up to 24 h)
[17] Cardinal et al., 2019	18 FM, 19 MDD, 29 HC (W)	spTMS: CSP, MEPs, RMT ppTMS: SICI, ICF	M1	VAS, NPS, HPT	BDNF, QST, BDI, PSQI, FIQ, STAI, B-PCS	FM (vs. MDD and HC): SICI↑	FM vs. MDD: BDNF↑, NPS→. r+ between NPS and SICI; r- between HPT and BDNF values	NR
[18] Deitos et al., 2018	17 FM + Pgb, 10 HC + Pgb (W)	spTMS: CSP, MEPs, RMT ppTMS: SICI, ICF	M1	VAS, NPS, PPT	BDI, B-PCS, STAI, MINI, FIQ, PSQI, BDNF, S100-B protein	FM + Pgb: VAS↓, SICI↓, CSP↑	FM + Pgb: r+ between BDNF and CSP; r- between S100-B and CSP	NR
[19] Schwenkreis et al., 2011	FMET: 16 FM, 23 MD, 23 HC. TMS before and after FMET	spTMS: CSP, RMT ppTMS: ICI, ICF	M1	None	None	FM and MD (pre-FMET): ICI↓. HC (post-FMET): ICI↓	-	NR
[20] Caumo et al., 2016	FM W (*n* = 19), MPS W (*n* = 54), OA W (*n* = 27), HC (*n* = 14)	spTMS: CSP, RMT, MEPs ppTMS: SICI, ICF	M1	CPM-NPS, QST	BDNF, BDI-II, B-PCS	FM and MPS: SICI↓	FM and MPS: r- between SICI and BDNF. r- between BDNF and NPS changes	NR

BDI-(II), Beck Depression Inventory; BDNF, brain-derived neurotrophic factor; B-PCS, Brazilian Portuguese version of the pain catastrophizing scale; CAT, catastrophizing; CPM, conditioned pain modulation test; CSP, cortical silent period; ERD, event-related desynchronization; ESS, Epworth sleepiness scale; FIQ, Fibromyalgia Impact Questionnaire; FM, fibromyalgia; FMET, fatiguing muscle exercise task; HC, healthy subjects control group; HPT, heat pain threshold; ICF, intracortical facilitation; ICI, intracortical inhibition; M1, primary motor cortex; MD, muscular dystrophy; MDD, major depressive disorder; MEPs, motor evoked potentials; MINI, Mini-international neuropsychiatric interview; MMSI, Mini-Mental State Examination; MPS, myofascial pain syndrome; NPS, numerical pain scale; NR, not reported; OA, osteoarthritis; PFC, pain-free controls; Pgb; pregabalin; pp, paired-pulse; PPT, pressure pain threshold; PSQI, Pittsburgh sleep quality index; QST, Quantitative sensory testing; r+, positive correlation; r- negative correlation RA, rheumatoid arthritis; RMT, resting motor threshold; RT, reaction time; SICI, short interval intracortical inhibition; sp, single-pulse; STAI, state-trait anxiety inventory; VAS, visual analog scale for pain; W, women; →, no significant changes; ↑, increased values; ↓, decreased values.

**Table 2 biomedicines-11-00954-t002:** Protocols and results of relevant studies conducted to explore rTMS effects on fibromyalgia symptoms.

Reference	Study Design	Sample Size	rTMS Protocol	Target Area	Intensity	Stimuli/Session	No. of Sessions	Physiological Measures	Pain Measures	Other Measures	Physiological Outcomes	Pain Outcomes	Other Outcomes	Adverse Effects
[21] Izquierdo-Alventosa et al., 2021	RDB, SC, PG: TMS vs. PE vs. control (NI)	49 FM W	60 trains/ 5 s pulses; 10 Hz; 15 s interval/each train	M1	80% RMT	3000 pulses	10	-	VAS, PPT	FIQR, PhC, 6-MWT, HAD, BDI-II, PSS-10, SLS	-	TMS: PPT↑, VAS↓. PE: PPT↑	TMS; 6 MWT↑, HAD↓, BDI-II↓, PSS-10↓, PhC↑. PE: FIQR↓, PhC↑, PSS-10↓, BDI-II↓, HAD↓, 6 MWT↑	NR
[22] Guinot et al., 2021	RDB, SC, PG. rTMS vs. sham plus AT +PBE +RXT	37 FM (33 W)	active vs. sham rTMS, 10 Hz	M1	80% RMT	2000 pulses	16 (12 for RXT)	CF, autonomic responses	VAS	FIQ, BDI, PSQI, PCS, PGIC	No BG differences: CF↑	No BG differences: VAS↓, PCS↓	No BG differences: BDI↓, FIQ↓	None
[23] Fitzgibbon et al., 2018	RDB, SC, PG	26 FM (24 W): 14 rTMS (13 W), 12 sham (11 W)	75 trains-4 s, 10 Hz	DLPFC	120% RMT	3000 pulses	20	-	SF-MPQ, BPI, NPRS	ACR, MFI, FIQ, PCS, PGIC, BDI-II, BAI	-	No BG differences: SF-MPQ↓, BPI↓, NPRS↓	rTMS: MFI↓	Discomfort, headaches, pain, nausea, dizziness (both groups)
[24] Short et al., 2011	RDB, SC, PG	20 FM (17 W): TMS (*n* = 10), sham (*n* = 10)	80 trains, 15 s, 10 Hz	DLPFC	120% RMT	4000 pulses	10	-	NPRS, BPI, FIQ	HDRS	-	Pre vs. post-TMS: NPRS↓, BPI↓	Pre vs. post-TMS: HDRS↓	Headache (2 subjects). No dropouts
[25] Mhalla et al., 2011	RDB, SC, PG	16 FM (rTMS), 14 FM (sham)	15 trains, 10 s pulses, 10 Hz, 50 s interval/train	M1	80% RMT	1500 pulses	14	RMT, MEPs, SICI, ICF	NPS, BPI, MPQ PCS	FIQ, HAD, BDI	Active rTMS: SICI↑, ICF↑. r- between SICI and NPS. r- between ICF and PCS, and between ICF and FIQ	Active rTMS: NPS↓, BPI↓, MPQ↓ (r+ FIQ). PCS↓	Active rTMS: FIQ↓	1 sham + 1 active dropped out (headache). Transient mild headache, dizziness (both groups)
[26] Tzabazis et al., 2013	DB, SC, Cr (4 coils rTMS vs. sham)	16 FM (14 W), 16 HC (11 W)	1/10 Hz vs. sham: PMF (×4 coils)	dACC	110% RMT	1800 pulses	1 (HC), 20 (FM)	-	BPI, NRS	-	-	1 Hz rTMS (HC): NRS↓. 10 Hz rTMS (FM): NRS↓	-	Pruritus, headache, back pain, neck pain, otalgia, nausea, lightheadedness, hot flashes, scalp pain (both groups)
[27] Forogh et al., 2021	RSB; rTMS vs. anodal tDCS effects	15 FM W (tDCS), 15 FM W (rTMS)	10 Hz, rest time (15 s)	DLPFC	100% RMT	1000 pulses	3	-	VAS	DASS-21, FIQR	-	rTMS and tDCS: VAS↓. rTMS effect > tDCS at 6 and 12 weeks of follow-up	-	Mild, transient, self-limiting headache (2 patients)

ACR, American College of Rheumatology Fibromyalgia Scale; AT, aerobic training; BAI, Beck Anxiety Inventory; BDI-(II), Beck Depression Inventory; BG, between groups; BPI, Brief Pain Inventory; CAT, catastrophizing; CF, cardiorespiratory fitness; Cr, crossover study; dACC, dorsal anterior cingulate cortex; DASS-21, the Depression Anxiety Stress Scale-21 items; DB, double-blind study; DLPFC, dorsolateral prefrontal cortex; FIQ, Fibromyalgia Impact Questionnaire; FIQR, Revised Fibromyalgia Impact Questionnaire; FM, fibromyalgia; HC, healthy subjects control group; HAD, Hospital Anxiety and Depression Scale; HDRS, Hamilton Depression Rating Scale; ICF, intracortical facilitation; M1, primary motor cortex; MEPs, motor evoked potentials; MFI, Multidimensional Fatigue Inventory-20; MPQ, McGill Pain Questionnaire; (6)-MWT, 6-Minute Walking Test; NI, no intervention; NPS, numerical pain scale; NPRS, Numerical Pain Rating Scale measurements of pain intensity and pain unpleasantness; NR, not reported; NRS, Numeric rating scale; PBE, pool-based exercises; PCS, Pain Catastrophizing Scale; PE, physical exercise; PG, parallel group; PGIC, personal global improvement of change; PhC, physical conditioning; PMF, pulsed magnetic fields; PPT, pressure pain threshold; PSQI, Pittsburgh sleep quality index; PSS-10, Perceived Stress Scale-10; r+ positive correlation; r- negative correlation; RDB, Randomized, double-blind study; RMT, resting motor threshold; RSB, Randomized, single blind study; rTMS, repetitive transcranial magnetic stimulation; RXT, relaxation therapy; SC, sham-controlled; SF-MPQ, Short-Form McGill Pain Questionnaire; SICI, short interval intracortical inhibition; SLS, Satisfaction with Life Scale; sp, single-pulse; VAS, visual analog scale for pain; W, women.

**Table 3 biomedicines-11-00954-t003:** Transcranial direct current stimulation (tDCS) protocols and results of relevant studies conducted to explore the neuromodulation effect on fibromyalgia symptoms.

Reference	Study Design	Sample Size	Anode electrode EEG Position	Return Electrode EEG Position	Target Area	Intensity	Electrode Size	Duration	Sessions Per Week	Total Sessions	Physiological Measures	Pain Measures	Other Measures	Physiological Outcomes	Pain Outcomes	Other Measures Outcomes	Adverse Effects
[27] Forogh et al., 2021	RSB, PG: tDCS vs. rTMS	15 (tDCS) + 15 (rTMS) FM W	F3	Fp2	DLPFC	2 mA	35 cm^2^	20 min	3	3	-	VAS	DASS-21, FIQR	-	rTMS and tDCS: VAS↓. rTMS effect > tDCS at 6 and 12 weeks of follow-up	-	Headache up to 24 h
[28] Morin et al., 2021	Pre vs. post tDCS + FT (single group, no sham)	10 FM W	C3/C4	Fp2/Fp1	M1 CL dominant hand	2 mA	35 cm^2^	20 min	5	5	-	VAS	Fatigue (Borg scale), FT	-	VAS→	FT↑	None
[29] de Melo et al., 2020	RDB, SC, PG	FM W: tDCS 5 days (11); tDCS 10 days (9); sham 5 days (11)	NS	Fp2	M1	2 mA	35 cm^2^	20 min	5 vs. 10	5 vs. 10	EEG oscillations (α frequency band) in F3, F4, P3, P4, O1, O2	VAS	CIRS, MMSE, BAI	tDCS (5 days): α↓ in frontal and parietal regions	All groups: VAS↓	-	None
[30] Lim et al., 2022	Cr, SC, SB: tDCS vs. sham	12 FM W and 15 HC W	C3	Fp2	M1	2 mA	35 cm^2^	20 min	5	5	fMRI BOLD	VAS, MPQ	-	FM baseline (vs. HC): BOLD↓ (vmPFC, lateral PFC, AI), BOLD↑ (PI). FM tDCS (vs. sham): BOLD↑ (rACC, lateral PFC, thalamus). Sham (vs. baseline): BOLD↓ (dorsomedial PFC, pCC, precuneus)	FM tDCS (vs. sham): VAS↓, r- between VAS and BOLD (rACC/vmPFC), r+ between VAS and BOLD (PI)	-	NR
[31] Yoo et al., 2018	RSB, SC, PG: DLPFC tDCS + ON tDCS vs. sham ON tDCS vs. ON tDCS, (sequential order)	58 FM: DLPFC + ON tDCS (21, 20 W); Sham ON tDCS (16, 15 W); ON tDCS (21, 20 W)	Anode: right DLPFC (NS EEG position) vs. anode: right ON	Left DLPFC vs. right ON	DLPFC vs. ON	2 mA (DLPFC), 1.5 mA (ON)	35 cm^2^	ON tDCS: 20 min, DLPFC + ON tDCS: 20 + 20 min	2	8	-	NRS	FIQ, BDI	-	ON tDCS: NRS↓	ON tDCS: FIQ↓, BDI↓. DLPFC + ON tDCS: BDI↓	Tingling, itching
[32] Samartin-Veiga et al., 2022	RDB, SC	M1 (29), DLPFC (26), OIC (28), sham (25) FM W	C3 vs. F3 vs. C5 (sham with any of these targets)	M1 and DLPFC: Fp2. OIC: F3, FC1, F8, FC5, P3	M1 vs. DLPFC vs. OIC	M1 and DLPFC: 2 mA. OIC: anode current: 1.144 mA	25 cm^2^ (M1 and DLPFC). 3.14 cm^2^ (OIC)	20 min	5	15	-	-	FSQ, SF-36, FIQR	-	-	All groups: SF-36↓, FIQR↓	9 dropouts (due to NR adverse effects)
[33] Kang et al., 2020	OS: pre- vs. post-tDCS	46 FM (44 W)	C3	C4	M1	2 mA	NR	20 min	5	5	-	VAS, BPI	FIQ, BFI, BDI, STAI, MOS-SS	-	Post-tDCS: VAS↓	Post-tDCS: FIQ↓, BDI↓, BFI↓	Mild dizziness, light headache, transient sleep disturbance. No dropouts
[34] Brietzke et al., 2020	RDB, SC, PG (in home, neoprene cap, tDCS)	10 FM W (anodal tDCS) + 10 FM W (sham)	F3	F4	DLPFC	2 mA	35 cm^2^	30 min	5	60	BDNF	VAS, B-PCP-S, PPT, HPT	Analgesic use, BDI, PSQI	tDCS: BDNF↑ predicted VAS↓	tDCS: VAS↓, analgesic use↓, B-PCP-S↓, HPT↑	tDCS: BDI↓, PSQI↓	Both groups: burning, tingling, itchiness, redness, headache, neck pain, mood swings, concentration difficulties
[35] de Paula et al., 2022	RDB, SC, PG: Active/sham tDCS vs. LDN (4.5 mg/day) vs. LDN placebo	FM W: LDN + tDCS (21), LDN + sham (22), LDN placebo + tDCS (22), LDN placebo + sham (21)	NS	CL SO area (NS EEG position)	M1	2 mA	NR	20 min	5	5	BDNF	VAS, PCS, PCP-S, PPT, CPM	STAI, FIQ, BDI-II	LDN + sham: BDNF↓. LDN placebo + tDCS: BDNF↓	LDN + tDCS, LDN + sham, placebo + sham: VAS↓. LDN + tDCS: PCP-S↓	LDN + tDCS: FIQ↓, STAI↓. All active interventions: BDI-II↓	tDCS: ↑rate of tingling, itching, blushing vs. sham. All tDCS groups: headache, neck ache, scalp pain, burning, sleepiness, mood changes
[36] To et al., 2017	RSB, SC, PG	42 FM (36 W): 15 C2; 11 DLPFC; 16 sham (8 C2 + 8 DLPFC)	F3 Left C2 nerve dermatome	F4 Right C2 nerve dermatome	DLPFC, ON	1.5 mA	35 cm^2^	20 min	2	8	-	NRS, PCS	MFIS	-	C2: NRS↓, DLPFC: NRS↓	DLPFC: MFIS↓	None
[37] Villamar et al., 2013	DB, SC, Cr (HD-tDCS 4 × 1 rings)	18 FM (15 W)	Center electrode: C3 (anode) vs. C3 (cathode) vs. sham	Cz, F3, T7, and P3	M1	2 mA	Ag/AgCl electrodes	20 min	1	3	-	VAS, SWMs (pain), PPT, DNICs	QOL, VAS (anxiety), BDI-II, SWMs (touch detection threshold)	-	Anodal/ cathodal: VAS↓	Anodal: SWMs↑	Active and sham: mild/moderate tingling, itching sensations (few min long)
[38] Desbiens et al., 2020	CS: sham + PhT (17 days later: tDCS + PhT)	1 FM W	Left M1 (NS EEG position)	NS	M1	2 mA	35 cm^2^	20 min	3	3	RMT	NRS, BPI	FIQ, NRS (fatigue), PhT performance	RMT→	tDCS + PhT: NRS↓, BPI↓	tDCS/ sham +PhT: NRS (fatigue)↓, PhT↑, FIQ↓	NR
[39] Valle et al., 2009	RDB, SC, PG	41 FM W: M1 = 14, DLPFC = 13, Sham M1 = 14	C3 vs. F3	Fp2	M1, DLPFC	2 mA	35 cm^2^	20 min	5	10	-	VAS	FIQ, BDI, IDATE, GDS, MMSE	-	M1 and DLPFC tDCS: VAS↓ (longer lasting effects with M1 tDCS)	M1 and DLPFC tDCS: FIQ↓	All groups: Skin redness, tingling
[40] Fagerlund et al., 2015	RDB, SC, PG	48 FM (45 W): tDCS: 24, sham: 24	C3	Fp2	M1	2 mA	35 cm^2^	20 min	5	5	-	NRS	FIQ, HADS, SCL-90R, SF-36	-	tDCS: NRS↓	tDCS: FIQ↓	Skin redness, sleepiness, tingling (no BG differences)
[41] Caumo et al., 2022	RDB, SC, PG (home based tDCS)	FM W: tDCS (30), sham (15)	F3	F4	DLPFC	2 mA	35 cm^2^	20 min	5	20	BDNF	NPS, PCS, PCP-S, HPT, HPTo	BDI-II, PSQI, FIQ, STAI, CSI	r+ between BDNF and PCP-S. r-between BDNF and PCS	tDCS: PCS↓, PCP-S↓, HPTo↑. r+ between PCS and PCP-S	tDCS: BDI-II↓, PSQI↓. r+ between PCP-S and BDI-II and PSQI	Tingling, burning, redness, headache, neck pain, mood swings, concentration difficulties (higher frequency in tDCS)
[42] Khedr et al., 2017	RDB, SC, PG	36 FM; tDCS: 18 (17 W), sham: 18 (17 W)	C3	CL arm	M1	2 mA	24 cm^2^	20 min	5	10	Beta-endorphin	VAS, WPI, PPT	SS, HAM-D, HAM-A	tDCS: r- between beta-endorphin and WPI and VAS	tDCS: VAS↓, WPI↓, PPT↓	tDCS: SS↓, HAM-A↓, HAM-D↓. r- between beta-endorphin and SS, HAM-A, and HAM-D	Itching and skin redness in 3 patients (tDCS group)
[43] Mendonca et al., 2011	RDB, SC, PG	30 FM (28 W) randomly divided into 5 groups	Anodal/cathodal C3 vs. anodal/cathodal Fp2 vs. sham	Transition of the cervical and thoracic spine	M1 vs. SO region	2 mA	80 cm^2^ extracephalic electrode, 16 cm^2^ cranial electrodes	20 min	1	1	E-field simulation	VNS, PPT, BD	-	Dominantly temporoparietal current flow in M1 configurations	Cathodal/anodal SO: VNS↓	-	Sham and real tDCS: mild tingling
[44] Silva et al., 2017	RDB, SC, Cr: tDCS vs. sham + a go/no-go task	40 FM W	F3	Fp2	DLPFC	1 mA	35 cm^2^	20 min	1 +1 (tDCS vs. sham)	1 +1 (tDCS vs. sham)	-	VAS, B-PCS, HPT, HPTo	PSQI, FIQ, BDI-II, MINI, ANT	-	tDCS: HPT↑, HPTo↑	tDCS: ANT↑	Tingling, burning, itching (no BG differences)
[45] Santos et al., 2018	RDB, SC, PG: tDCS + DN-B	FM W: tDCS (19), sham (20)	F3	Fp2	DLPFC	2 mA	35 cm^2^	20 min	NR	8	BDNF	-	DN-B, RAVLT, PASAT, COWAT, FDS, BDS	tDCS: r- between BDNF and RAVLT	-	tDCS: RAVLT↑, COWAT↑, FDS↑	NR
[46] De Ridder et al., 2017	RDB, SC, Cr: tDCS vs. sham (after washout)	19 FM (15 W) and 19 HC	Right C2 dermatome	Left C2 dermatome	OCF	1.5 mA	35 cm^2^	20 min	3 active tDCS + 3 sham	3 active tDCS + 3 sham	sLORETA, EEG	NRS, PCS	FIQ	FM baseline: dorsal ACC↑, PM/DLPFC↑; after tDCS, pregenual ACC↑	tDCS: NRS↓, PCS↓	tDCS: FIQ↓	Transient redness and slight itching after tDCS
[47] Foerster et al., 2015	L, Cr, NRa: tDCS vs. sham (after washout)	12 FM W	Left M1 (NS EEG position)	Right SO (NS EEG position)	M1	2 mA	NR	20 min	5 active tDCS + 5 sham	5 active tDCS + 5 sham	^1^H-MRS	VAS, LF-MPQ, SF-MPQ	PANAS	tDCS: Glx (ACC)↓, sham: NAA (PI)↑, tDCS/sham: r- between baseline Glx (ACC) and VAS after stimulation	tDCS: VAS↓	tDCS: PANAS↓	NR
[48] Matias et al., 2022	RDB, SC, PG: tDCS + FE	31 FM W: tDCS + FE (17), sham + FE (14)	C3	Fp2	M1	2 mA	35 cm^2^	20 min	5	5	-	VAS, PPT	MWT6, SSt, FIQ, BDI, MINI, VAS (anxiety), MFI, FS	-	Real and sham tDCS+ FE: VAS↓, PPT↑	-	Headache, tingling, dizziness, nausea (no BG differences)
[49] Samartin-Veiga et al., 2022	RDB, SC, PG	130 FM W: M1 tDCS: 34, DLPFC: 33, OIC: 33, sham: 30	C3 vs. F3. OIC with multielectrode montage: FC5: 0.579 mA and C5: 1.144 mA	M1 and DLPFC: Fp2. OIC: F3: −0.565 mA, FC1: −0.508 mA, F8: −0.158 mA, and P3: −0.492 mA	M1, DLPFC, OIC	2 mA	25 cm^2^ (M1/DLPFC), 3.14 cm^2^ disc electrodes (OIC)	20 min	5	15	-	NRS, FIQ, PPT	FIQ (fatigue), HADS, MFE-30, PSQI	-	All groups: NRS↓, PPT↑, FIQ↓	All groups: MFE-30↓, PSQI↓, FIQ↓. tDCS groups: HADS↓	Tickling, itching, burning. No BG differences
[50] Castillo-Saavedra et al., 2016	OL, single arm, phase II (HD-tDCS; no sham)	14 FM (12 W)	C3	CZ, T7, P3 and F3	M1	2 mA	Standard Ag/AgCl ring electrodes	20 min	5	≤26	EEG	VAS, PPT, SWMs	FIQ, BDI	Responders: baseline BNA↑	VAS↓, VAS (50%)↓ in 7 out of 14 patients	FIQ↓	Tingling sensation (scalp), mild headache, mild pain, skin redness
[51] Cummiford et al., 2016	SB, Cr, NRa: tDCS vs. sham (after washout)	12 FM W	C3	Fp2	M1	2 mA	35 cm^2^	20 min	5 for sham tDCS (one week apart, 5 for active tDCS)	5 sham + 5 active tDCS	rsFC fMRI	VAS, SF-MPQ	PANAS	sham: rsFC↓ (VPL-SI-Am). tDCS: rsFC↓ (VLT, mPFC, SMA)	tDCS and sham: r- between rsFC (M1-VLT, S1-AI, VLT-PAG) and VAS. tDCS and sham: r+ between rsFC (VLT/VPL-PI, M1, S1) and VAS	tDCS: PANAS negative affect↓	NR
[52] Roizenblatt et al., 2007	RDB, SC, PG	32 FM W: Sham (10), M1 tDCS (11), DLPFC tDCS (11)	C3 vs. F3 vs. sham C3	Fp2	M1, DLPFC	2 mA	35 cm^2^	20 min	5	5	PSG	VAS	VAS (tiredness, anxiety), CGI, PGA, BDI, FIQ, SF-36, MMSE	M1 tDCS: SE↑, SA↓; r- between BM and SE, r- between SE and FIQ, r+ between REM latency and FIQ, r+ between SL and VAS, r+ between SL and FIQ. DLPFC tDCS: SE↓, REM↑, SL↑	M1 tDCS: VAS↓	M1 tDCS: FIQ↓	Well-tolerated; no BG differences
[53] Riberto et al., 2011	RDB, SC, PG: tDCS + multiple rehabilitation	FM W: 11 tDCS + 12 sham	C3	Fp2	M1	2 mA	35 cm^2^	20 min	1	10	-	VAS, SF-36 (pain)	FIQ, HAQ, BDI, HAM-D	-	SF-36↓ (larger in the tDCS group)	No BG differences	None
[54] Fregni et al., 2006	RDB, SC, PG	32 FM W (M1 tDCS: 11; DLPF tDCS: 11; sham: 10)	C3 vs. F3	Fp2	M1, DLPFC	2 mA	35 cm^2^	20 min	5	5	-	VAS	CGI, PGA, FIQ, SF-36, BDI, VAS (anxiety), MMSE, Stroop, DSfb, RT task	-	VAS↓ (greater in the M1 tDCS group)	M1 tDCS: FIQ↓, SF-36↑	All groups: sleepiness and headache
[55] Plazier et al., 2015	RDB, SC, Cr: OCF subcutaneous stimulation, then ON tDCS and sham	9 FM W	Left C2 dermatome	Right C2 dermatome	OCF	1.5 mA	35 cm^2^	20 min	3	3	-	NRS	-	-	ON tDCS: NRS↓. r+ between NRS (ON tDCS) and short-term NRS (invasive stimulation)	-	None
[56] Mendonca et al., 2016	RDB, SC, PG: tDCS + AE	45 FM (44 W); M1 tDCS + AE: 15; Sham + AE: 15; M1 tDCS: 15	C3	Fp2	M1	2 mA	35 cm^2^	20 min	5	5	ICI, ICF	VNS, PPT	VNS (anxiety), SF-36, BDI	No BG differences	M1 tDCS + AE group (vs. M1 tDCS alone): VAS↓	VNS (anxiety)↓, BDI↓ (in both real tDCS groups but larger in the tDCS + AE)	Tingling, skin redness (no BG differences)
[57] DalĺAgnol et al., 2015	CS: M1 tDCS vs. DLPFC tDCS vs. sham	1 FM W	NR	NR	M1, DLPFC	2 mA	NR	20 min	NR	10 each intervention	-	VAS, PCS	FIQ, Brazilian STAI, BDI	-	DLPFC tDCS: VAS↓, PCS↓; M1 tDCS: VAS↓	DLPFC: STAI (trait)↓, RC↓; M1: STAI (state)↓, BDI↓	Skin redness and tingling
[58] Ramasawmy et al., 2022	RDB, SC, PG: tDCS + MM vs. sham + MM vs. NI	30 FM (28 W); M1 tDCS + MM: 10; sham + MM: 10; NI: 10	5 cm to the left of Cz	Right SO area (NS EEG position)	M1	2 mA	Anode: 16 cm^2^ Cathode: 50 cm^2^	20 min	5	10	-	NRS, PPT	FIQ, DASS-21, NRS (sleep quality)	-	No BG differences	tDCS + MM: FIQ↓	Light headache, vertigo, fatigue, nervousness, skin redness (no BG differences)
[59] Serrano et al., 2022	RDB, SC, PG: Home-based tDCS	36 FM W: DLPFC: 24; sham: 12	F3	F4	DLPFC	2 mA	35 cm^2^	20 min	5	20	BDNF	CPM, PPT, B-PCS	TMT, WAIS-III (Ds), COWAT, FIQ, BDI-II, PSQI	r- between BDNF and TMT, Ds, and COWAT. r+ between BDNF and FIQ	Pain͢͢͢͢͢ measures͢͢→	DLPFC tDCS: TMT↑, Ds↑, COWAT↑, FIQ↓	Headache, tingling, burning, redness, itching (no BG differences)
[60] Arroyo-Fernández et al., 2022	RDB, SC, PG: anodal vs. sham tDCS + exercising, vs. NI)	120 FM (113 W)	C3	Fp2	M1	2 mA	25 cm^2^	20 min	3 + 2	5	-	VAS, PPT	FIQ, IDATE, PCS, BDI-II	-	Pre vs. post anodal tDCS: VAS↓	Pre vs. post anodal and sham tDCS: PCS↓, BDI-II↓, FIQ↓	None
[61] La Rocca et al., 2022	RDB, SC, PG: anodal vs. sham tDCS +FTT	54 FM (41 W, anodal tDCS: 28; sham: 26); 22 HC (16 W, anodal tDCS: 11; sham: 11)	C3	Fp2	M1	2 mA	35 cm^2^	20 min	1	1	fNIRS	-	FTT	Anodal tDCS: M1 activation↑	-	FTT no BG differences	NR

ACC, anterior cingulate cortex; AE, aerobic exercise; AI, anterior insula; Am, amygdala; ANT, Attention Network Test; BAI, Beck Anxiety Inventory; BD, body diagram evaluating pain area; BDI (BDI-II), Beck Depression Inventory; BDNF, brain-derived neurotrophic factor; BDS, Backward Digit Span; BFI, Brief Fatigue Inventory; BG; between group; BM, body mass; BNA, pain-related brain network activation; BOLD; Blood-oxygenation-level-dependent; B-PCP-S, Brazilian Portuguese version of the Profile of Chronic Pain-Screen; B-PCS, Brazilian Portuguese version of the pain catastrophizing scale; BPI, Brief Pain Inventory; CGI, Clinician global impression; CIRS, Cumulative Illness Rating Scale; CL, contralateral; COWAT, Controlled Oral Word Association Test; CPM, conditioned pain modulation test; Cr, crossover study; CS, case study; CSI, Central Sensitization Inventory; DASS-21, the Depression Anxiety Stress Scale-21 items; DB, double-blind study; DLPFC, dorsolateral prefrontal cortex; DN-B; Dual N-Back task; DNICs, diffuse noxious inhibitory controls; Ds; digits subtest; DSfb, Digit span forward and backward; EEG, electroencephalography; ecs, electrode configurations; ei, each intervention; FDS, forward digit span; FE, functional exercise; FIQ, Fibromyalgia Impact Questionnaire; FIQR, Revised Fibromyalgia Impact Questionnaire; FM, fibromyalgia; FMI, Freiburg Mindfulness Inventory; fMRI, functional magnetic resonance imaging; fNIRS, Functional Near Infrared Spectroscopy; FS, Feeling Scale; FSQ, Fibromyalgia Survey Questionnaire; FT, number of repetitions in a functional task; FTT, finger-tapping task; GDS, Geriatric Depression Scale; Glx, glutamate + glutamine; HADS, Hospital Anxiety and Depression Scale, HAM-A, Hamilton Anxiety Scale; HAM-D, Hamilton Depression Scale; HAQ, Health Assessment Questionnaire; ^1^H-MRS, proton magnetic resonance spectroscopy; HC, healthy subjects control group; HPT, heat pain threshold; HPTo, heat pain tolerance; ICF, intracortical facilitation; ICI, intracortical inhibition; IDATE, state-trait anxiety inventory for anxiety; M1, primary motor cortex; L, longitudinal study; LDN, low-dose oral naltrexone; LF-MPQ, Long-Form McGill Pain Questionnaire; MF, mobile phone (message); MFE-30, Memory Failures of Everyday Questionnaire; MFI, Multidimensional Fatigue Inventory-20; MFIS, Modified Fatigue Impact Scale; MINI, Mini-International Neuropsychiatric Interview; MM, mindfulness meditation; MMSE, Mini-Mental State Examination; MOS-SS, Medical Outcomes Study Sleep Scale; mPFC, medial prefrontal cortex; MPQ, McGill Pain Questionnaire; MWT6, 6-min walk test; NAA, N-acetylaspartate; NI, no intervention; NR, not reported; NRa, not randomized study; NRS, Numeric rating scale for pain; NS, not specified; OCF, occipital nerve field; OIC, the operculo-insular cortex; OL, open-label study; ON, occipital nerve; OS, open study; PAG; periaqueductal gray; PANAS, positive and negative affect scores; PASAT, Paced Auditory Serial Addiction Test; pCC, posterior cingulate cortex; PCP-S, Profile of Chronic Pain Scale; PCS, Pain Catastrophizing Scale; PFC, prefrontal cortex; PG, parallel group; PGA, Patient Global Assessment; PhT, physical task; PI, posterior insula; PM, premotor cortex; PPT, pressure pain threshold; PSG, polysomnography; PSQI, Pittsburgh sleep quality index; QOL, quality of life; r+, positive correlation; r-, negative correlation; rACC, rostral anterior cingulate cortex; RAVLT, Rey Auditory-Verbal Learning Test; RC, ruminative catastrophism; RMT, resting motor threshold; RT, reaction time; RDB, Randomized, double-blind study; RL, randomized, longitudinal study; RSB, Randomized, single blind study; rsFC, resting state functional connectivity; rTMS, repetitive transcranial magnetic stimulation; SA, sleep arousal; SB, single-blind; SC, sham-controlled; SCL-90R, Symptom Checklist 90; SE, sleep efficiency; SF-36, 36-item short-form health survey questionnaire; SF-MPQ, Short-Form McGill Pain Questionnaire; SI, primary somatosensory cortex; SL, sleep latency; sLORETA; low-resolution brain electromagnetic tomography; SMA, supplementary motor area; SO, supraorbital cortex; SS, symptom severity of fibromyalgia; SSt; sit-to-stand test; STAI, state-trait anxiety inventory; SWMs, Semmes-Weinstein monofilaments (mechanical and pain detection threshold); TMT, Trail Making Test; VAS, visual analog scale for pain; VLT, ventral lateral thalamus; vmPFC, ventromedial prefrontal cortex; VNS, Visual Numeric Scale; VPL, ventral posterolateral thalamus; W, women; WAIS-III, Wechsler Adult Intelligence Scale; WMT, working memory training; WPI, widespread pain index; →, no significant changes; ↑, increased values; ↓, decreased values

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
