# Peer review of "Non-Invasive Electric and Magnetic Brain Stimulation for the Treatment of Fibromyalgia"

_biomedicines, 2023, doi:10.3390/biomedicines11030954_

Round 1
Reviewer 1 Report
Thank you for the opportunity to review your work. This narrative review is well done and frankly hard to improve on. I do have some minor comments to consider.
1. The authors could consider a systematic review with focused questions and PRISMA guidance. This would increase the rigor and impact of the paper and I would imagine it would not take much additional effort.
2. Regardless a succinct statement regarding the purpose and goals of this narrative/unstructured review would be helpful. This could be added near the end of section 3.
3. It would be good to indicated that this a narrative review in the abstract and early in the manuscript.
4. A search strategy should be summarized. This can be brief but currently it is not clear how the articles were selected.
Author Response
Responses to the reviewers’ comments
Reviewer 1
- This narrative review is well done and frankly hard to improve on. I do have some minor comments to consider.
Response:
Thank you for your valuable comments. All changes made in response to the reviewer comments appear in red font in the new version of the manuscript, and we also refer to the page numbers where each change can be found.
- The authors could consider a systematic review with focused questions and PRISMA guidance. This would increase the rigor and impact of the paper and I would imagine it would not take much additional effort.
Response:
We are grateful to this reviewer for calling our attention to this issue. We have modified our previous narrative review so that the new version of the manuscript shows now a systematic review following the PRISMA criteria. This is stated in the abstract and the Introduction sections, and we have added a Method section according to PRISMA:
Page 3 (Introduction): “In this systematic review we focus on the involvement of the central nervous system in this syndrome and critically evaluate how non-invasive brain stimulation might be a useful method...”
Page 3 (Methods): “2. Methods. This review follows the Preferred Reporting Items for Systematic Reviews and Meta-Analyses (PRISMA) guidelines”
- Regardless a succinct statement regarding the purpose and goals of this narrative/unstructured review would be helpful. This could be added near the end of section 3.
Response:
We appreciate this comment. As we have adapted the manuscript to the PRISMA guidelines, now our review includes a Method section, in which Search strategy and selection criteria, and Data extraction and analysis sub-sections were added. In these sections we clarify the objectives of this review, the database search strategy, and the criteria to select articles:
Page 3 (Method section):
“2.1. Search strategy and selection criteria
Our search focused on research studies about neurophysiological alterations in fibromyalgia as evaluated by transcranial magnetic stimulation, as well as studies investigating the effects of non-invasive magnetic or electric brain stimulation methods on fibromyalgia symptoms. We conducted a systematic search in PubMed and Web of Science databases using the following keywords combinations…”
Page 4 (Method section):
“2.2. Data extraction and analysis
We analyzed the following information from the full-text articles: study population (fibromyalgia patients, patients with other clinical conditions, healthy controls), sample size (per group and total), study design (between-subjects with parallel groups, within subject-crossover, single case study, single- or double-blind, open-label study, randomized study, sham-controlled, longitudinal study), TMS-dependent neurophysiological measures of cortical excitability, TMS and tDCS protocols and stimulation parameters, cortical stimulation target, tDCS electrode position, total number of stimulation sessions and duration, sessions per week, pain measures, questionnaires and other outcome measures, tasks, physiological measures, results reported, and adverse effects”
- It would be good to indicated that this a narrative review in the abstract and early in the manuscript.
Response:
Many thanks for this suggestion. We have indicated in the abstract (as well as at the end of the Introduction section) that this is a systematic review:
Page 1 (Abstract): “…In this systematic review we outline current evidence showing the potential of non-invasive brain stimulation techniques as therapeutic strategy in fibromyalgia. In addition, we evaluate the contribution of these tools to the exploration of the neurophysiological characteristics of fibromyalgia…”
- A search strategy should be summarized. This can be brief but currently it is not clear how the articles were selected.
Response:
As this reviewer suggests, we have included a Search strategy and selection criteria sub-section in the Method section:
Page 3:
“2.1. Search strategy and selection criteria
…We conducted a systematic search in PubMed and Web of Science databases using the following keyword combinations: transcranial magnetic stimulation OR TMS OR rTMS AND fibromyalgia OR transcranial electric stimulation OR tES AND fibromyalgia OR transcranial direct current stimulation OR tDCS AND fibromyalgia OR transcranial alternating current stimulation OR tACS AND fibromyalgia OR transcranial random noise stimulation OR tRNS AND fibromyalgia OR transcutaneous vagus nerve stimulation OR tVNS AND fibromyalgia. The database search included publications from 2000 until January 2023. This search yielded a total of 572 results, as shown in the PRISMA flow diagram of Figure 1.
We selected research articles that met the following criteria: adult participants with fibromyalgia, studies with non-invasive electric and/or magnetic brain stimulation (open label, single or double blind, with or without sham condition and healthy participants, randomized or not randomized studies). The search was restricted to full-text original articles published in English. Fifty-four studies met these criteria and were included in this systematic review (Figure 1). Review and study protocol articles were only reported in the Discussion section or to support general concepts, but they were not considered as research results…”

Reviewer 2 Report
The work is certainly driven by an interesting idea, especially considering the controversial issues affecting the medical etio-pathogenesis of fibromyalgia.
The most concerning problem is the methodological one.
I am not quite sure if Biomedicine journal does accept narrative reviews. However, as it currently stands, the manuscript looks as though it were.
In other words, although the work of Molero-Chamizo et al. covered a topic of paramount relevance, both from a clinical and societal standpoint, the means by which the review was established is very difficult to determine. Stipulates that the review is not a narrative point of view, yet there is an extreme limit to the scope of material included within the review. Managing a complex multifactorial disease like fibromyalgia would suggest to conduct a systematic review in order to provide weighted statements and claims. It would be easier indeed to read conclusions and stipulations coming from a systematic review of literature rather than a narrative point of view. Therefore I would recommend to stipulate the question and/or hypothesis that you are investigating and indicate the terms used in developing your search for literature and the returns obtained from each search. Please may the Authors want to compile a dedicated "Methods" section.
Author Response
Reviewer 2
- The work is certainly driven by an interesting idea, especially considering the controversial issues affecting the medical etio-pathogenesis of fibromyalgia.
Response:
Thank you for your valuable comments. All changes made in response to the reviewer comments appear in red font in the new version of the manuscript, and we also refer to the page numbers where each change can be found.
- The most concerning problem is the methodological one. I am not quite sure if Biomedicine journal does accept narrative reviews. However, as it currently stands, the manuscript looks as though it were. In other words, although the work of Molero-Chamizo et al. covered a topic of paramount relevance, both from a clinical and societal standpoint, the means by which the review was established is very difficult to determine. Stipulates that the review is not a narrative point of view, yet there is an extreme limit to the scope of material included within the review. Managing a complex multifactorial disease like fibromyalgia would suggest to conduct a systematic review in order to provide weighted statements and claims. It would be easier indeed to read conclusions and stipulations coming from a systematic review of literature rather than a narrative point of view.
Response:
We are very grateful for this valuable comment. As this reviewer suggests, we have modified our previous narrative review so that the new version of the manuscript shows now a systematic review following the PRISMA guidance:
Page 3 (Methods): “2. Methods. This review follows the Preferred Reporting Items for Systematic Reviews and Meta-Analyses (PRISMA) guidelines”
- Therefore I would recommend to stipulate the question and/or hypothesis that you are investigating and indicate the terms used in developing your search for literature and the returns obtained from each search.
Response:
Thank you for this suggestion. We have specified the objective of this review:
Page 3 (Introduction): “In this systematic review we focus on the involvement of the central nervous system in this syndrome and critically evaluate how non-invasive brain stimulation might be a useful method..”
And we have indicated the terms used in our database search and the returns obtained:
Page 3 (Methods): “2.1. Search strategy and selection criteria. Our search focused on research studies about neurophysiological alterations in fibromyalgia as evaluated by transcranial magnetic stimulation, as well as studies investigating the effects of non-invasive magnetic or electric brain stimulation methods on fibromyalgia symptoms. We conducted a systematic search in PubMed and Web of Science databases using the following keywords combinations: transcranial magnetic stimulation OR TMS OR rTMS AND fibromyalgia OR transcranial electric stimulation OR tES AND fibromyalgia OR transcranial direct current stimulation OR tDCS AND fibromyalgia OR transcranial alternating current stimulation OR tACS AND fibromyalgia OR transcranial random noise stimulation OR tRNS AND fibromyalgia OR transcutaneous vagus nerve stimulation OR tVNS AND fibromyalgia. The database search included publications from 2000 until January 2023. This search yielded a total of 572 results, as shown in the PRISMA flow diagram of Figure 1.
We selected research articles that met the following criteria: adult participants with fibromyalgia, studies with non-invasive electric and/or magnetic brain stimulation (open label, single or double blind, with or without sham condition and healthy participants, randomized or not randomized studies). The search was restricted to full-text original articles published in English. Fifty-four studies met these criteria and were included in this systematic review (Figure 1). Review and study protocol articles were only reported in the Discussion section or to support general concepts, but they were not considered as research results”
- Please may the Authors want to compile a dedicated "Methods" section.
Response:
As this reviewer suggests, we have included a “Methods” section in the new version of the manuscript. This section includes the information supplied above, as well as a “Data extraction and analysis” sub-section:
Page 4 (Methods): “2.2. Data extraction and analysis. We analyzed the following information from the full-text articles: study population (fibromyalgia patients, patients with other clinical conditions, healthy controls), sample size (per group and total), study design (between-subjects with parallel groups, within subject-crossover, single case study, single- or double-blind, open-label study, randomized study, sham-controlled, longitudinal study), TMS-dependent neurophysiological measures of cortical excitability, TMS and tDCS protocols and stimulation parameters, cortical stimulation target, tDCS electrode position, total number of stimulation sessions and duration, sessions per week, pain measures, questionnaires and other outcome measures, tasks, physiological measures, results reported, and adverse effects”

Reviewer 3 Report
The article is interesting. I have several suggestions:
First, the prevalence of fibromyalgia in the general and disease-specific population should be addressed.
Second, a simple graph to summarize the session titled “Nervous system involvement in fibromyalgia” would be helpful for the readers.
Third, regarding rTMS for pain reduction, there are several review articles published recently. Please consider to reference:
https://pubmed.ncbi.nlm.nih.gov/36683849/
https://pubmed.ncbi.nlm.nih.gov/36836613/
Fourth, although this is a narrative review, it would be helpful for the readers to know which key words had been used to identify relevant articles.
Fifth, in the discussion, please add some subtitles to ease the reading.
Author Response
Reviewer 3
- The article is interesting. I have several suggestions
Response:
Thank you for your valuable comments. All changes made in response to the reviewer comments appear in red font in the new version of the manuscript, and we also refer to the page numbers where each change can be found.
- First, the prevalence of fibromyalgia in the general and disease-specific population should be addressed.
Response:
Following the suggestion of this reviewer, the Introduction section of the new version of the manuscript includes now a statement about the prevalence of fibromyalgia:
Page 1 (Introduction): “…Recent fibromyalgia prevalence studies estimate values between 0.2 and 6.6% in the general population, between 2.4 and 6.8 % in women, maximum values of 11.4% in urban areas, and 15% in special populations, such as car accident survivors [4]”
- Second, a simple graph to summarize the session titled “Nervous system involvement in fibromyalgia” would be helpful for the readers.
Response:
We agree with this comment, and, accordingly, we have included the following paragraph to summarize the section “Nervous system involvement in fibromyalgia”:
Page 6 (Nervous system involvement in fibromyalgia): “…In summary, evidence indicates that some fibromyalgia symptoms involve central nervous system mechanisms, mainly cortical plasticity alterations and central sensitization, and these perturbances of the nervous system may be tackled by non-invasive brain stimulation methods”
- Third, regarding rTMS for pain reduction, there are several review articles published recently. Please consider to reference:
https://pubmed.ncbi.nlm.nih.gov/36683849/
https://pubmed.ncbi.nlm.nih.gov/36836613/
Response:
We are grateful to this reviewer for providing the references of those reviews. We have included them in the new version of the manuscript:
Page 20 (Conclusions): “…Thus, rTMS protocols described in clinical fibromyalgia studies have shown similar effectiveness for relieving rheumatic [101,102] and post-stroke [103] pain…”
- Fourth, although this is a narrative review, it would be helpful for the readers to know which key words had been used to identify relevant articles.
Response:
We are grateful to this reviewer for calling our attention to this issue. As mentioned above, in the current systematic review we have included a “Methods” section, and a “Search strategy and selection criteria” sub-section, in which the search keywords were defined:
Page 3 (Methods): “…We conducted a systematic search in PubMed and Web of Science databases using the following keywords combinations: transcranial magnetic stimulation OR TMS OR rTMS AND fibromyalgia OR transcranial electric stimulation OR tES AND fibromyalgia OR transcranial direct current stimulation OR tDCS AND fibromyalgia OR transcranial alternating current stimulation OR tACS AND fibromyalgia OR transcranial random noise stimulation OR tRNS AND fibromyalgia OR transcutaneous vagus nerve stimulation OR tVNS AND fibromyalgia…”
- Fifth, in the discussion, please add some subtitles to ease the reading.
Response:
As this reviewer suggests, the Discussion section is now organized into the following sub-sections for easier reading:
4.1. TMS and cortical neurophysiology in fibromyalgia
4.2. Therapeutic effects of non-invasive brain stimulation on fibromyalgia symptoms
4.2.1. tDCS
4.2.2. rTMS
4.3. Future directions and limitations

Round 2
Reviewer 2 Report
I thank the Authors for having fully addressed my raised concerns.
On a singular note, the theoretical framework in the introduction would benefit by inserting papers with positive relevance of portable tDCS on fitness, such as:
doi: 10.1055/a-1198-8525
Thank you very much for the opportunity to review this manuscript.
Author Response
Many thanks for this suggestion. In the Introduction section (page 3), we have included the suggested reference as follows:
"...we highlight recent studies that point to the potential of non-invasive electric and magnetic brain stimulation [9,10] as a therapeutic approach to treat at least some of its neurological and neuroinflammatory symptoms [11], taking into account the apparent effectiveness of these methods to improve some physical indicators of health [12]"